# Gender Differences in the Frequency of Positive and Negative Effects after Acute Caffeine Consumption

**DOI:** 10.3390/nu15061318

**Published:** 2023-03-07

**Authors:** Przemysław Domaszewski

**Affiliations:** Department of Health Sciences, Institute of Health Sciences, University of Opole, 45-040 Opole, Poland; przemyslaw.domaszewski@uni.opole.pl

**Keywords:** side effects, positive effects, gender, caffeine, optimalisation

## Abstract

Gender-specific caffeine-related adverse effects should be thoroughly investigated. Sixty-five adult participants were included in the study, 30 men and 35 women (age, 22.5 ± 2.8; body weight, 71.7 ± 16.2 kg; BMI, 23.6 ± 4.4). The participants who were classified as low and moderate caffeine users received 3 mg/kg, and high caffeine users received 6 mg/kg of caffeine in one dose. One hour after ingestion of caffeine and within twenty-four hours, the participants completed a side effect questionnaire. Effects after the ingestion of CAF were divided into two subgroups: negative (muscle soreness, increased urine output, tachycardia and palpitations, anxiety or nervousness, headache, gastrointestinal problems, and insomnia) and positive (perception improvement; increased vigor/activeness). Caffeine ingestion resulted in a statistically significant association between gender and negative effects one hour after ingestion (*p* = 0.049). Gender and positive effects one hour after ingestion (*p* = 0.005), and between gender and positive effects within 24 h after ingestion (*p* = 0.047). There were significant associations between gender and perception improvement (*p* = 0.032) and gender and increased vigor/activeness (*p* = 0.009) one hour after ingestion. Nearly 30% of men and 54% of women reported negative effects. At the same time, 20% of women and more than 50% of men reported positive effects. Gender is an important factor in the negative and positive effects of caffeine consumption.

## 1. Introduction

Caffeine (CAF) is the most commonly used stimulant of the nervous system and a substance that enhances athletic performance and cognitive function [1,2]. After ingestion, caffeine readily crosses biological membranes, and maximum plasma concentrations are reached within 30 to 40 min. The elimination half-life of caffeine in the blood is usually about 3 to 7 h. The main mechanism of action of caffeine is its antagonistic effect on adenosine receptors [3]. Adenosine is an essential cellular component that functions as an extracellular signaling molecule. The binding of caffeine molecules to adenosine receptors stimulates the sympathetic part of the autonomic nervous system, reduces fatigue, and increases concentration [4]. The impact of caffeine on a parasympathetic branch of the autonomic nervous system is less extensively investigated [3]. Another mechanism of CAF is the inhibition of phosphodiesterase (PDE), which leads to an increased concentration of cyclic adenosine monophosphate and the inhibition of γ-aminobutyric acid receptors [5]. At low doses of CAF, the basic mode of action in the central nervous system is binding to adenosine receptors [2]. However, at higher doses of caffeine, many other molecular targets, such as γ-aminobutyric acid receptors, may also play an important role [6]. Caffeine increases sympathetic nerve outflow, circulating catecholamine concentration, plasma renin activity, and blood pressure. Caffeine, through the adenosine receptors, may also modulate dopamine secretion. The suggested mechanisms involving the regulation of dopamine synthesis or the rate of dopaminergic neuron firing rate may be mainly responsible for affecting behavioral activation, mood, cognition, and effort-related [7].

The Food Drugs and Administration recommends a dose of 400 mg of CAF per day, not associated with hazards or side effects for the general population [8]. The recommended dose is often exceeded—especially since regular caffeine users become habituated to the caffeine, and even high doses can be ineffective in this case [9]. However, in very high doses, this xanthine alkaloid can interact strongly with the sympathetic nervous system and cause serious adverse effects on the cardiovascular system [2]. Usually, the recommended and maximum amounts of caffeine are given in absolute numbers. It is stated how much coffee can be consumed without health risks. This approach seems to be wrong. First, different types of coffee and their blends contain different amounts of caffeine. In addition, the caffeine content of the beverage changes depending on the method of preparation, and the number of variables that affect caffeine content is very high, e.g., brewing time, brewing quantity, degree of grinding, and tamping. For example, the caffeine content of an espresso in the same coffee house can vary from 75.8 mg to 140.4 mg [10]. There are also weighty reasons for refraining from stating caffeine doses in absolute values, e.g., 200 mg per day, since differences in body weight certainly affect the concentration of caffeine in the organism and thus the positive or negative effects. The latest research clearly indicates that body weight-based dosing should also be modified, especially in athletes, and that lean mass content should be the basic parameter for evaluating the appropriate caffeine dose that provides maximum benefit and minimum risk of side effects [11].

Considering the body weight factor, recommended doses of CAF vary but can be divided into three groups: low (3 mg per kg body weight), moderate (between 5 and 6 mg per kg body weight), and high (≥9 mg per kg body weight [12]. Doses up to 6 mg per kg body weight have been shown to have no serious adverse effects in individuals who are not hypersensitive to caffeine [13,14,15,16,17,18]. Higher doses (≥9 mg per kg body weight) of caffeine may cause adverse effects, such as decreased reaction time, gastrointestinal problems, dizziness, insomnia, nervousness, irritability, and poor concentration [17,19,20,21,22]. On the contrary, high doses of caffeine may also be ineffective in invigorating habitual CAF users [16], and athletes who typically consume large amounts of caffeine require higher doses to enhance their performance [19]. On the other hand, the International Society of Sports Nutrition indicates that higher doses (≥9 mg per kg body weight) do not enhance performance [17] and may overstimulate the central nervous system, leading to acute symptoms such as headaches, anxiety, insomnia, muscle tremors, increases in blood pressure, and changes in cardiac rhythm [20,22]. However, there is a possibility that minor adverse effects may occur, and individuals who are hypersensitive to caffeine may experience side effects even at lower doses [23]. Doses greater than 2000 mg per day pose a real health risk and can cause hypertension, cardiac arrhythmias, convulsions, and even death [24,25,26].

Following caffeine consumption, people’s responses vary, with a range of negative, neutral, or positive effects depending on genotype, age, habitual caffeine consumption, training status, CAF source, and gender [27]. The exact mechanism underlying these differences remains unknown, but body size and body composition could be crucial factors. Temple and Ziegler’s findings also suggest that responses to caffeine may be related to changes in steroid hormone levels. In women, higher estradiol levels may be associated with lower or no response to CAF, while lower estradiol levels are associated with a negative response to CAF compared with placebo [28]. Women also appear more likely to experience breathing problems, palpitations, hand tremors, and adverse gastrointestinal symptoms following the use of CAF [29,30]. However, only a few studies have examined gender alternations in the side effects of CAF [31]. It has been suggested that individual differences in fat and lean body mass contribute to differences in caffeine absorption, occurrence rates, and side effects [32,33]. CAF is essentially distributed only in lean body mass, so the higher ratio of adipose tissue to lean body mass may result in higher plasma and tissue concentrations of CAF and greater stimulant effects in overweight and obese individuals when the CAF dose is calculated in mg per kg of total body weight [11,34].

Despite a wealth of evidence on the effects of CAF on physical and cognitive functions, these findings are difficult to implement in practice. This is primarily because there are few studies that explicitly consider gender as a factor influencing caffeine intake [35]. Identifying components that have the potential to influence the uptake of CAF and the magnitude of side effects may help to optimize caffeine dosing while minimizing the occurrence of side effects [11]. The hypothesis of the study is that there are gender differences in the positive and negative effects of acute caffeine consumption.

## 2. Materials and Methods

### 2.1. Study Participants

Sixty-five adult participants aged 19 to 29 years, with 30 men (age = 24.5 ± 3.8; body weight = 79.5 ± 13 kg, height = 182 ± 5.1 cm; BMI = 23.84 ± 3.3) and 35 women (age = 22.7 ± 2.7; body weight = 64.9 ± 15.7 kg, height = 166.6 ± 6.4 cm; BMI = 23.4 ± 5.19), were included in the experiment. To ensure safety, the participants received different doses of caffeine depending on the amount of daily caffeine consumption. Individuals with low and moderate daily caffeine intake received 3 mg CAF per kg body weight (n = 34; 14 men and 19 women), and individuals with high daily caffeine intake received 6 mg CAF per kg body weight (n = 31; 16 men and 15 women).

Inclusion criteria:

Age 18–30 years old;Absence of medical contraindications;No hypersensitivity to caffeine;Consent to participate in the study and statement of compliance with the guidelines;Absence of electronic life support systems (pacemakers, active prostheses, etc.).

### 2.2. Caffeine Intake Assessment

Using the Food Frequency Questionnaire (FFQ) by Bühler et al. [36], thirty-one participants were classified as high consumers of CAF (351 ± 139 mg caffeine per day), nineteen as moderate, and fifteen as low caffeine consumers [37]. Based on household portions, the amount of food CAF consumed was classified according to the following frequency of consumption: more than three times daily, two to three times daily, once daily, five to six times weekly, two to four times weekly, once weekly, three times monthly, rarely, or never. A list of foods with high to moderate CAF content was provided, including coffee, energy drinks, and other popular beverages containing CAF, green and black tea, cocoa products, CAF supplements, and medications.

### 2.3. Experiment Design

Depending on daily caffeine intake, the participants were divided into 2 groups: CAF3, n = 33, and CAF6, n = 32 (3 and 6 mg caffeine per kg body weight). Pure caffeine was administered in a dose calculated on body weight in transparent cellulose capsules. The capsules were washed down with water. The trial participants were asked not to change their CAF consumption habits before the day of the experiment. Since caffeine is usually taken in the morning and early afternoon, all of the measurements were taken between 7:00 a.m. and 3:00 p.m., at least 2 h after a light meal. On the day of the experiment and for the following 24 h, the participants were asked to avoid products containing caffeine.

### 2.4. Negative and Positive Effects after Caffeine Ingestion Questionnaire (QUEST)

One hour after the ingestion of caffeine and after twenty-four hours, the participants answered a standardized side effect questionnaire (QUEST) designed to evaluate the positive and negative effects of caffeine [16,38,39], containing nine items (muscle soreness, increased urine output, tachycardia and heart palpitations, anxiety or nervousness, headache, gastrointestinal problems, insomnia, perception improvement, and increased vigor/activeness) rated on a yes/no scale.

### 2.5. Side Effects

The effects after the ingestion of CAF were divided into 2 subgroups: negative effects (muscle soreness, increased urine output, tachycardia and heart palpitations, anxiety or nervousness, headache, gastrointestinal problems, and insomnia) and positive effects (perception improvement; increased vigor/activeness).

### 2.6. Ethics

All of the participants in the study signed an informed consent form. The study was approved by the Bioethics Committee of the Medical University of Poznań (No. 108/22), registered in the Australian–New Zealand Clinical Trials Registry (No. 12622000823774), and conducted according to the guidelines for research involving human participants described in the Declaration of Helsinki. All of the tests were performed in the physiological laboratory of the Medical University of Poznań.

### 2.7. Statistical Analysis

To determine the relationships between the variables, the authors used Pearson’s chi-square test for independence with continuity correction, and Cramer’s V was used to determine the relative measure (strength) of the relationship between the two variables. StatsCloud software (https://statscloud.app/beta/) (accessed on 10 February 2023) was used to perform the statistical analyzes and descriptive statistics. The size of the intervention group was analyzed using the G * Power 3.1.9.2 program. The total sample size of the 65 participants in each of the two groups resulted in an effect size of (0.35) and a significance level of 5%.

## 3. Results

Table 1 shows the results of the statistically significant associations between the GENDER and POSITIVE/NEGATIVE effects that occurred 60 min and 24 h after the ingestion of CAF. The results of the study show a statistically significant association between GENDER and NEGATIVE effect 60 min after the ingestion of caffeine (Χ^2^ = 3.89, *p* = 0.049, Cramer’s V = 0.24). The results also show a statistical association between GENDER and POSITIVE effect 60 min after caffeine ingestion (Χ^2^ = 7.85, *p* = 0.005, Cramer’s V = 0.35). Within 24 h of caffeine ingestion, a statistically significant association was found between GENDER and POSITIVE effect (Χ^2^ = 3.96, *p* = 0.047, Cramer’s V = 0.22). The association between GENDER and NEGATIVE within 24 h after caffeine ingestion was not statistically significant. Table 2 shows the associations between the GENDER predictor and certain NEGATIVE effects 60 min after CAF ingestion. Table 3 shows the associations between the GENDER predictor and certain POSITIVE effects 60 min after CAF ingestion. Table 4 shows the associations between the GENDER predictor and certain NEGATIVE effects 24 h after CAF ingestion, and Table 5 shows the associations between the GENDER predictor and certain POSITIVE effects 24 h after CAF ingestion. The results in Table 2, Table 3, Table 4 and Table 5 show a statistically significant association between GENDER and perception improvement (Χ^2^ = 4.59, *p* = 0.032, Cramer’s V = 0.27) 60 min after CAF ingestion and GENDER and increased vigor/activeness (Χ^2^ = 6.81, *p* = 0.009, Cramer’s V = 0.32) 60 min after CAF ingestion. The exact dose- and sex-dependent number of participants who reported specific adverse events immediately after the test protocol and 24 h later is shown in Table 6.

## 4. Discussion

This study is one of the first attempts to analyze gender differences in positive and negative effects after acute caffeine ingestion. The novelty of this study is the use of doses that depend on the daily caffeine intake of the participants. The study showed a significant correlation between gender and adverse effect 60 min after the ingestion of caffeine. Nearly 30% of men suffered a negative effect when a dose of 3 mg per kg body weight was used in the low and moderate caffeine user group and 6 mg per kg body weight in the high caffeine user group. At the same time, 54% of the participants in the female group reported at least one of the adverse effects. The study also showed a significant association between gender and beneficial effects within one hour of caffeine ingestion. One in five female participants reported positive effects from caffeine. In the male group, more than 50% of the participants reported positive effects. There were no significant differences between gender/negative or positive effects 24 h after caffeine ingestion. Detailed analysis of the results revealed a significant association between gender and perceptual improvement and between gender and increased vigor/activeness shortly after caffeine ingestion. One in three men and only one in ten women reported perceptual improvement shortly after caffeine ingestion (35% and 11%, respectively). A similar pattern emerged in the analysis of increased vigor or activity. One hour after caffeine ingestion, 43% of men and 14% of women reported increased vigor or activeness.

When analyzing the dose- and sex-dependent number of participants reporting adverse effects one hour after the testing protocol and 24 h later, increased urine excretion was one of the most commonly reported adverse effects. The effect of increased urine excretion after doses of 3 and 6 mg per kg body weight occurred in both the female and male groups; however, this effect was observed mainly in females. The participants reported increased urine secretion primarily shortly after ingestion, although a small effect on 24 h urine secretion was observed in the male group. The sex differences in increased urine secretion have already been described in studies by Massey [34] and Wu [40], and the fact of increased urine secretion immediately after caffeine consumption is physiological. The mechanism is related to a transient increase in blood pressure leading to increased glomerular filtration in the kidneys [41]. The blood pressure-increasing effect of caffeine is well established [42], and the results observed in the literature suggest that CAF causes a significant increase in blood pressure, especially in naïve individuals. In contrast, in habitual caffeine users, the effect of adenosine receptor adaptation leads to a lack of response to caffeine intake with an increase in blood pressure [2]. The effect of caffeine on heart rate has been associated with a baroreflex adaptation that attempts to compensate for the increase in blood pressure [43].

The frequency of the occurrence of anxiety or nervousness was more than three times higher in the female group than in the male group, and these results are consistent with the findings of Pez-Graniel [44]. Although some animal studies suggest that caffeine has the potential to reduce anxiety [45], the authors found no such effect in this study, but this may be related to the relatively low doses of caffeine used in the study. The lowest dose used by Sweene was 6 mg per kg body weight, and the highest was up to 24 mg per kg body weight. Studies of the effect of higher doses of caffeine on anxiety in humans are sparse but appear promising in terms of long-term reduction of anxiety. The beneficial effects of caffeine on cognitive functions, including improving perception and increasing vigor/activity, are well known and demonstrated [46,47]. These effects were particularly noted in the group of men shortly after caffeine ingestion and were dose-independent. Although there is a wealth of evidence on the effects of caffeine on physical and cognitive functions, there are few studies examining positive and negative effects as a function of gender. In particular, for athletes who regularly take caffeine supplements, factors related to side effects following caffeine consumption are important because adverse effects could be critical to athletic performance. Among professional athletes, the use of caffeine is widespread, and more than 92% of athletes take caffeine supplements or drink coffee, especially before competitions or training [35]. Supplementation with caffeine and the use of energy drinks by athletes to increase ergogenic effects may lead to adaptation, and therefore the doses used in sports are usually higher than the recommended doses. With sports supplements, including pre-training blends and caffeine capsules or powders, doses equivalent to 5 caffeine units (e.g., 600 mg in one dose) or more can easily be ingested. In a previous study, we found that a dose of 9 mg CAF per kg body weight was required to produce direct positive effects on skeletal muscle mechanical activity in professional handball players who regularly consumed caffeinated products [2]. Thus, in sports, the question is not whether caffeine is ergogenic at a particular dose but whether the dose taken provides optimal performance benefits without adverse effects [48,49].

Considering the latest studies of Skinner et al. and Surma et al. indicating that caffeine is mainly distributed through free body mass [11,20], the greater proportion of adipose tissue may result in higher plasma and tissue concentrations of caffeine. If women tend to have relatively less free fat mass and more fat mass than men [50,51], this will also influence caffeine absorption, differences in plasma concentrations, and the occurrence of side effects [32]. The authors suggested that this fact is responsible for the significant differences between the sexes and the more negative effects in women compared to the male group after caffeine intake. In addition, men with a high percentage of adipose tissue could suffer from negative effects after caffeine intake if the caffeine dose is administered considering body weight. Extensive research should be conducted to determine the optimal caffeine dose considering the percentage of muscle mass rather than body weight.

## 5. Conclusions

When low (3 mg/kg body weight) and moderate (6 mg kg body weight) doses of caffeine were used, 30% of men and 54% of women suffered an adverse effect. The frequency of the occurrence of anxiety or nervousness was more than three times higher in the female group than in the male group. Almost 20% of the female and more than 50% of male participants reported positive effects after caffeine intake. Significant differences in body composition, particularly free fat mass content, are likely responsible for these differences.

## 6. Limitations

The author did not investigate the negative and positive effects of ingesting high doses (over 6 mg per kg body weight) of CAF and did not determine the concentration of caffeine and its metabolites in the blood. In addition, the authors did not analyze serum steroid hormones, which could be one of the factors for the gender differences in the occurrence of side effects.

## Figures and Tables

**Table 1 nutrients-15-01318-t001:** Association between GENDER predictor and NEGATIVE/POSITIVE effects after CAF ingestion.

Outcome	Predictor	Χ^2^	*p*	Cramers V
Negative effect 60 min after CAF Yes/No	Gender	3.86	0.049	0.24
Positive Effect 60 min after CAF Yes/No		7.85	0.005	0.35
Negative effect 24 h after CAF Yes/No		2.65	0.104	0.20
Positive Effect 24 h after CAF Yes/No		3.96	0.047	0.25

**Table 2 nutrients-15-01318-t002:** Association between Gender predictor and particular NEGATIVE effects 60 min after CAF ingestion.

Outcome	Predictor	Χ^2^	*p*	Cramers V
Muscle soreness *	Gender	-	-	-
Increased urine output		0.23	0.632	0.06
Tachycardia and heart palpitations		0.63	0.429	0.1
Anxiety or nervousness		1.79	0.181	0.17
Headache		0.21	0.648	0.06
Gastrointestinal problems		0.11	0.744	0.04
Insomnia *		-	-	-

* not observed (the muscle soreness 60 min after CAF intake variables has 1 levels but require 2).

**Table 3 nutrients-15-01318-t003:** Association between Gender predictor and certain POSITIVE effects 60 min after CAF ingestion.

Outcome	Predictor	Χ^2^	*p*	Cramers V
Perception improvement	Gender	4.59	0.032	0.27
Increased vigor/activeness		6.81	0.009	0.32

**Table 4 nutrients-15-01318-t004:** Association between Gender predictor and certain NEGATIVE effects 24 h after CAF ingestion.

Outcome	Predictor	Χ^2^	*p*	Cramers V
Muscle soreness	Gender	0.78	0.376	0.11
Increased urine output		2.405	0.121	0.19
Tachycardia and heart palpitations		1.92	0.166	0.17
Anxiety or nervousness		0.006	0.938	0.01
Headache		0.41	0.525	0.08
Gastrointestinal problems		0.006	0.938	0.01
Insomnia		1.92	0.166	0.17

**Table 5 nutrients-15-01318-t005:** Association between Gender predictor and certain POSITIVE effects 24 h after CAF ingestion.

Outcome	Predictor	Χ^2^	*p*	Cramers V
Perception improvement	Gender	3.25	0.072	0.22
Increased vigor/activeness		0.564	0.453	0.09

**Table 6 nutrients-15-01318-t006:** The dose and sex-dependent number of participants who reported side effects immediately after the test protocol and 24 h later.

Side Effects	Group	Men (n = 30)		Women (n = 35)	
		60 min	24 h	60 min	24 h
Muscle soreness	CAF3	0 (0%)	0 (0%)	0 (0%)	0 (0%)
	CAF6	0 (0%)	0 (0%)	0 (0%)	0 (0%)
Increased urine output	CAF3	2 (7%)	2 (7%)	6 (17%)	0 (0%)
	CAF6	5 (17%)	1 (3%)	4 (11%)	0 (0%)
Tachycardia and heart palpitations	CAF3	0 (0%)	0 (0%)	1 (3%)	3 (9%)
	CAF6	0 (0%)	0 (0%)	0 (0%)	0 (0%)
Anxiety or nervousness	CAF3	0 (0%)	0 (0%)	5 (14%)	0 (0%)
	CAF6	1 (3%)	1 (3%)	1 (3%)	1 (3%)
Headache	CAF3	0 (0%)	0 (0%)	2 (6%)	0 (0%)
	CAF6	3 (10%)	1 (3%)	0 (0%)	3 (9%)
Gastrointestinal problems	CAF3	0 (0%)	0 (0%)	2 (6%)	1 (3%)
	CAF6	1 (3%)	1 (3%)	0 (0%)	0 (0%)
Insomnia	CAF3	0 (0%)	0 (0%)	0 (0%)	3 (9%)
	CAF6	0 (0%)	0 (0%)	0 (0%)	0 (0%)
Perception improvement	CAF3	4 (13%)	1 (3%)	1 (3%)	0 (0%)
	CAF6	6 (20%)	3 (10%)	3 (9%)	0 (0%)
Increased vigor/ activeness	CAF3	8 (27%)	3 (10%)	1 (3%)	2 (6%)
	CAF6	5 (17%)	1 (3%)	4 (11%)	0 (0%)

Data are presented as the number of participants (frequency) that responded affirmatively to the existence of a side effect. CAF3—caffeine 3 mg per kg body weight group (men, n = 14; women, n = 20); CAF6—caffeine 6 mg per kg body weight group (men, n = 16; women, n = 15).

## Data Availability

The datasets generated and/or analyzed during the current study are not publicly available. However, the data are available from the corresponding author upon reasonable request.

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
