# Peer review of "Gender Differences in the Frequency of Positive and Negative Effects after Acute Caffeine Consumption"

_nutrients, 2023, doi:10.3390/nu15061318_

Round 1

Reviewer 1 Report

The study is interesting, gender differences are important in terms of caffeine effects. there are 49 papers cited in the text, while in the list of References one can find only 41 items.

Most of the cited research is relevant. 

Author Response

Please find enclosed the revised version of the manuscript written by Przemysław Domaszewski, which I submitted for publication. I have taken into account all the points raised by the reviewers. All changes in the revised version of the manuscript are marked in yellow.
I hope you find it suitable for publication in Nutrients.

Yours respectfully,

Przemysław Domaszewski

Reviewer 1. The study is interesting, gender differences are important in terms of caffeine effects. there are 49 papers cited in the text, while in the list of References one can find only 41 items.

Dear Reviewer, I'm grateful for your insightful comments on my manuscript. I've carefully reviewed and corrected all errors related to the bibliography.

Reviewer 2 Report

·         Replace the word “Subject” with “Participant” throughout the text.

Introduction

·         Lines 47-48: Provide 3-5 references for the following sentence:

“The Food Drugs and Administration recommends a dose of 400 mg of CAF per day, not associated with hazards or side effects for the general population.”

·         Lines 48-52: Provide reference(s) for the following sentences:

“The recommended dose is often exceeded - especially since regular caffeine users become habituated to the caffeine and even high doses can be ineffective in this case. However, in very high doses, this xanthine alkaloid can interact strongly with the sympathetic nervous system and cause serious adverse effects on the cardiovascular system.”

·         The study hypothesis must be added at the end of “Introduction”.

Methods

·         Day-to-day test reliability, CV range, and intraclass correlation coefficients for the assessments must be included for ALL the assessments.

·         Suggestion: Add a schematic representation of the study procedures to the “Methods” section.

Discussion

·         The "Discussion" section must clarify the study's novelty better.

·         The limitations of this study have been adequately elaborated.

Author Response

Please find enclosed the revised version of the manuscript written by Przemysław Domaszewski, which I submitted for publication. I have taken into account all the points raised by the reviewers. All changes in the revised version of the manuscript are marked in yellow.
I hope you find it suitable for publication in Nutrients.

Yours respectfully,

Przemysław Domaszewski

Reviewer 2.

  • Replace the word “Subject” with “Participant” throughout the text.

Dear Reviewer, I'm grateful for your suggestion about my manuscript. I've replaced the word "Subject" with "Participant" throughout the text.

Introduction

  • Lines 47-48: Provide 3-5 references for the following sentence:

“The Food Drugs and Administration recommends a dose of 400 mg of CAF per day, not associated with hazards or side effects for the general population.” 

Thank you very much for your suggestion. I have added reference no. 8: Panel, E.; Nda, A. Scientific Opinion on the safety of caffeine. EFSA J. 2015, 13, doi:10.2903/j.efsa.2015.4102.

  • Lines 48-52: Provide reference(s) for the following sentences:

“The recommended dose is often exceeded - especially since regular caffeine users become habituated to the caffeine and even high doses can be ineffective in this case. However, in very high doses, this xanthine alkaloid can interact strongly with the sympathetic nervous system and cause serious adverse effects on the cardiovascular system.”

Thank you very much for your suggestion. I have added reference of Santangelo (2018) and Domaszewski (2021).

  • The study hypothesis must be added at the end of “Introduction”.

I added the study hypothesis as suggested:

The hypothesis of the study is that there are gender differences in the positive and negative effects after acute caffeine consumption.

Methods

  • Day-to-day test reliability, CV range, and intraclass correlation coefficients for the assessments must be included for ALL the assessments.

Thank you very much for your suggestion. I have added the paragraph detailing the questionnaire with references to studies using the same method:

2.4. Negative and positive effects after caffeine ingestion questionnaire (QUEST), with reference to three papers that used QUEST questionnaires to assess the effects of caffeine (Wilk, et.al: The effects of high doses of caffeine on maximal strength and muscular endurance in athletes habituated to caffeine; Desbrow and Leveritt: Awareness and use of caffeine by athletes competing at the 2005 Ironman Triathlon World Championships and Childs and deWit: Subjective, behavioral, and physiological effects of acute caffeine in light, nondependent caffeine users). In my study, I used the questionnaire established in other publications, so I did not analyze the reliability of the day-to-day test, the CV range, and the intraclass correlation coefficients in my manuscript. Furthermore, the aim of my study was not to validate the questionnaire already used in previous work.

  • Suggestion: Add a schematic representation of the study procedures to the “Methods” section.

I appreciate your suggestion, however, I believe that all procedures are clearly described.

Discussion

  • The "Discussion" section must clarify the study's novelty better.

Thank You for Your suggestion. I added to the discussion section:

This study is one of the first attempts to analyze gender differences in positive and negative effects after acute caffeine ingestion. The novelty of this study is the use of doses that depend on the daily caffeine intake of the participants.